# Algorithms for Automatic Data Validation and Performance Assessment of MOX Gas Sensor Data Using Time Series Analysis

**Christof Hammer** [1,2,*], **Sebastian Sporrer** [2], **Johannes Warmer** [1], **Peter Kaul** [1], **Ronald Thoelen** [3] and **Norbert Jung** [1]

1   Institute of Safety and Security Research ISF, University of Applied Sciences Bonn-Rhine-Sieg, Grantham Allee 20, 53757 Sankt Augustin, Germany
2   Institute for the Protection of Terrestrial Infrastructures, German Aerospace Center, Rathaus Allee 12, 53757 Sankt Augustin, Germany
3   Institute for Materials Research, Hasselt University, Wetenschapspark 1, B-3590 Diepenbeek, Belgium
*   Correspondence: christof.hammer@dlr.de; Tel.: +49-2241-20148-31

**Abstract:** The following work presents algorithms for semi-automatic validation, feature extraction and ranking of time series measurements acquired from MOX gas sensors. Semi-automatic measurement validation is accomplished by extending established curve similarity algorithms with a slope-based signature calculation. Furthermore, a feature-based ranking metric is introduced. It allows for individual prioritization of each feature and can be used to find the best performing sensors regarding multiple research questions. Finally, the functionality of the algorithms, as well as the developed software suite, are demonstrated with an exemplary scenario, illustrating how to find the most power-efficient MOX gas sensor in a data set collected during an extensive screening consisting of 16,320 measurements, all taken with different sensors at various temperatures and analytes.

**Keywords:** time series analysis; MOX gas sensors; slope based signature; automatic measurement validation; prioritizable ranking; feature extraction

## 1. Introduction

Since metal-oxide (MOX) gas sensors are cheap, easy to acquire and available in large quantities, they have become popular in different measurement scenarios such as leakage detection in chemical factories or air quality measurements in central venting systems [1,2]. The sensors can detect gas concentrations down to the ppb level, but suffer from the disadvantage of not being selective enough. Hence, researchers continuously create and test new material combinations with the goal of building sensors that are very selective and sensitive to a specific target [3]. In addition to the actual composition of the sensitive layer, the sintering parameters used for the process of applying the metal-oxide onto the empty sensor carrier impacts the sensor's performance immensely. Therefore, custom-made sensors are manufactured in batches with the same metal oxide composition, but individual sinter parameters. In order to test the achieved individual sensitivity and selectivity of the sensors in a batch, all sensors are exposed simultaneously but sequentially to different gases whilst being operated at different substrate temperatures. This procedure is called a sensor screening [4,5].

Depending on the granularity, a screening can be a very time-consuming task (i.e., several days) and should ideally be highly automatized. In our previous work, we presented hardware solutions for automated batch sintering [6] and a sensor readout system to carry out highly automated sensor screenings [7]. Since the acquired data has to be analyzed and interpreted to achieve the final goal of finding the best fitting sensor and its optimal operating temperature for a given target, an automated measurement hardware

for sensor screenings is only half way to the goal. Due to the large amount of raw data captured during an automated screening with many parameter combinations, a manual interpretation can also be very time-consuming and will therefore benefit greatly from a high degree of automation itself.

There are two main challenges identified for the automatized processing and analysis of the data acquired during an automated screening that have to be addressed algorithmically:

- Validation
  Since manufacturing and operating parameters are still under research, some sensors may show a malformed or no response at all. The occurrence of such invalid measurements in a screening is therefore very likely. These measurements need to be sorted out, to only include measurements from proper sensors for the final assessment.
- Ranking
  To identify the best sensor for a given application, a performance metric is required. It should be based on quantifiable and individually prioritizable features extracted from the time series measurements. The ability to tune the metric through feature-wise prioritization will help to model the scenario, for which the ranking is performed, in greater detail.

In the following work, we will present method combinations and algorithms needed to address the identified challenges. To find invalid or unusual measurements, we propose an automatic validation method based on curve similarity that compares new measurements against a well-known reference to determine how correlative they are. An algorithm calculates a numeric similarity value between the given reference and the curves under test. A threshold can then be used to automatically sort out measurements that are too dissimilar from the reference. Furthermore, we propose a signature extraction algorithm that significantly enhances the performance of the well-established curve metrics, directly improving the numeric similarity results. The solution for the sensor ranking is split into feature extraction and the ranking algorithm itself. Sensor-expert interviews led to the identification of several MOX sensor-specific features. A relevant set of features, extractable from the sensors' time series, was mathematically formalized. Finally, we can use the resulting feature vectors as input for our proposed ranking algorithm, which is based on multiplicative arithmetic. The ranking can individually prioritize a freely selectable combination of features from the vector, to best possibly adapt the ranking process to the target application for the sensor.

We will conclude by showing the developed algorithms and software on an exemplary ranking performed on a data set obtained during an extensive sensor screening, to automatically find the most sensitive sensor for a given analyte, while consuming as little power for its heater element as possible.

## 2. Related Work and Data Origin

Since this works primary contributions are algorithms and methods for automatic data validation and ranking of newly manufactured sensors for application of specific detection tasks, we looked at similar work in this field.

Many research groups like Leo et al. [8] mention their data processing as using several individual pieces of heterogeneous, commercial software tools like LabView or Matlab, or scripts. Often, the used sensor-features and statistical methods are presented informally as incomplete textual expressions or as black boxes entirely [9]. This makes a reproduction of the algorithms difficult. The software Dave3 [10], is a toolbox with a graphical user interface, which comes close to the idea, we want to convey. The tool is, however, specialized for the evaluation of data obtained during temperature cyclic operation of gas sensors and not applicable for the validation and performance ranking for sensors according to their screening data. Another unnamed software for the evaluation of data obtained from an electronic nose could be found. The software is limited to a specific subset of sensors and also is not suitable for ranking or validating data [11]. Both tools have

the major shortcoming of being based on commercial software like MatLab or LabView which requires additional licences.

Our goal is to present the algorithms for validation and ranking as well as the needed features in a mathematically formalized way so that can they can be implemented in a variety of (open source) languages of choice, such as Python or R.

### 2.1. Sensor Screening Method

The data used in this work is the result of a detailed screening of 64 sensors exposed to nine different analytes. The sensors under test differ in their substrate composition as well as their sinter times and sinter temperatures used during their production. As the operating temperature greatly impacts the sensor performance, a single physical sensor operated at different temperatures can be regarded as multiple virtual sensors with very different sensitivity and selectivity [12]. The operating temperatures were therefore varied during the screening, to record the resulting impact on the sensor performance. Sensor resistance, heater voltage and heater current were sampled with 1 Hz during the entire screening. A single measurement for a sensor and an analyte at a given temperature is repeated at least three times before the temperature is changed and the cycle starts over. The resulting time series for each measurement in the described data set always consists of the following three segments and durations as defined by the screening procedure:

Figure 1 is an exemplary depiction of the result from a single measurement. The dotted vertical lines indicate the analyte exposure to the sensor, while the toned down color of the curve, left and right of the dotted lines, is used to visualize the baseline and clearing segments as described in Table 1.

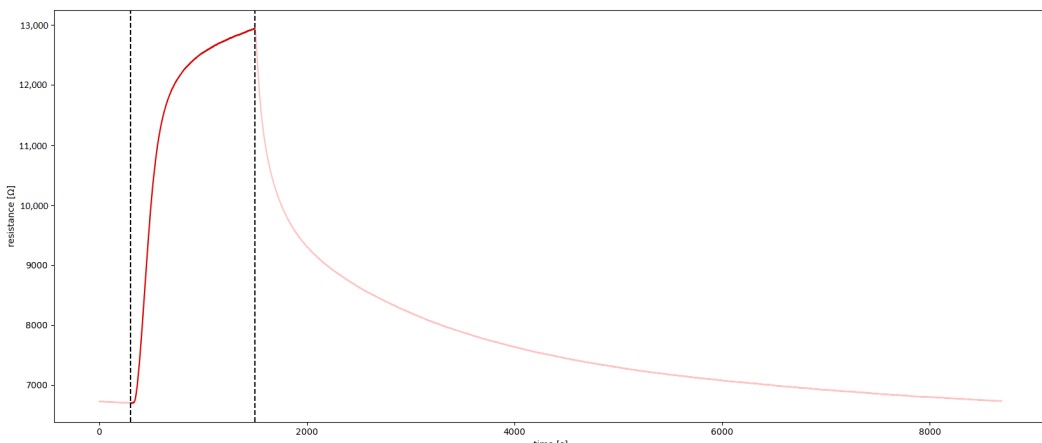

**Figure 1.** Measurement of a $Cr_2O_3$ sensor, sintered with $800\,^{\circ}C$ for 720 min, operated at $450\,^{\circ}C$, exposed to 5 ppm Acetone. The section between the dotted lines is the analyte exposition. The slightly toned down color before and after the analyte are the baseline and clearing parts of the measurement.

**Table 1.** Basic structure of an individual measurement.

| Segment | Duration | Action |
|---------|----------|--------|
| **(B)** Baseline | 5 min | Get sensor value in synthetic air before analyte |
| **(A)** Analyte | 20 min | Get sensor value during gas exposition |
| **(C)** Clearing | 120 min | Flush sensor and piping with synthetic air |

### 2.2. Preprocessing

An out of range (OOR) detection algorithm checks that each sample lies within a range defined by a fixed lower boundary of $0\ \Omega$ and a customizable upper boundary $\tau_r$, whereas $\tau_r$ ideally coincides with the maximal measurable resistance value of the measurement equipment. If more than 10 consecutive samples are outside these boundaries, the measurement is flagged as invalid.

If optional information for heater voltage and current is available, it can be used to detect and remove measurements that were performed with sensors that presumably have broken or malfunctioning heater elements. These extended checks are:

- Regulation Deviation
  The recorded heater voltage is compared to the targeted voltage. The algorithm counts the occurrences of deviations of $\pm 5\%$ to the target voltage. If this occurs more than 10 times, the measurement is flagged as erroneous.
- Continuous Current Flow
  Checks that the current is actually flowing through the heater element throughout the entirety of the measurement.
- Heater-Characteristic
  Using the parameters from the technical information bulletin provided by UST Umweltsensortechnik GmbH [13], the resistance characteristic of the integrated platinum heater element of the sensors at different temperatures can be validated. Sensors that were exposed to long sintering times at high temperatures are especially prone to damage to their heater element. Such sensors can be flagged with a warning.

The system presented in our previous work [7] provides this additional data and was used for all screenings. Therefore, the extended preprocessing is applied to all measurements in the available data set. According to the upper limit of the used measurement system, the upper boundary for the range checks is $\tau_r = 4\,\text{G}\Omega$. Finally, an outlier detection was performed to correct for single-sample signal anomalies.

## 3. Algorithms for Validation and Ranking

Based on the challenges described in Section 1, the following solutions are proposed:

- Slope-based signature calculation as additional curve similarity metric to enhance a distance-based, semi-automatic measurement validation process;
- Feature-based and prioritizable ranking metric to sort the sensors according to their performance towards a given analyte.

Before getting started, the formal conventions are introduced. In this work a vector is denoted with $\mathbf{x} \in \mathbb{R}^{|\mathbf{x}|}$ where $|\mathbf{x}|$ is defined as the amount of the vector's elements. The vector element at index $i$ is referenced by $\mathbf{x}[i]$ with $1 \leqslant i \leqslant |\mathbf{x}|$. A window with size $w \in \mathbb{N}$ around an index $i$ can be interpreted as a vector itself containing only values from the original vector $\mathbf{x}$ with the limits $l \leqslant i \leqslant r$.

$$\mathbf{x}\langle i, w\rangle \quad \text{with} \quad l = \begin{cases} i - w & \text{if } i - w > 1, \\ 1 & \text{else} \end{cases} \quad \text{and} \quad r = \begin{cases} i + w & \text{if } i + w < |\mathbf{x}|, \\ |\mathbf{x}| & \text{else} \end{cases} \tag{1}$$

Let $\mathbf{v} = (1 \quad 2 \quad 3 \quad 4 \quad 5)$ be a vector with five elements. An exemplary window could then be $\mathbf{v}\langle 3, 1\rangle = (2 \quad 3 \quad 4)$. Furthermore, the last $n$ elements of the vector could also be addressed via the window $\mathbf{v}\langle |\mathbf{v}|, n - 1\rangle$. For $n = 3$, this gives $\mathbf{v}\langle 5, 2\rangle = (3 \quad 4 \quad 5)$.

A measurement is a vector $\mathbf{m} \in M$ with $M$ being the set of all resistance measurements of one or multiple sensor screenings, as introduced in Section 2.1. The elements of this vector with their corresponding indices represent the sampled values and the time base of the measurement. Following this notation, the different segments of a measurement are defined as $\mathbf{m}_a$ for the analyte exposure, $\mathbf{m}_b$ for the baseline and $\mathbf{m}_c$ for the clearing phase.

The residual standard deviation used in the work is defined as [14]:

$$s(\mathbf{x}, \mathbf{r}) = \sqrt{\frac{\sum_{i=1}^{|\mathbf{x}|} (\mathbf{x}[i] - \mathbf{r}[i])^2}{|\mathbf{x}| - 2}}, \ |\mathbf{x}| > |\mathbf{r}| \tag{2}$$

All notation conventions, including those introduced above, are summarized in Table 2 as a compact overview.

**Table 2.** Summary of notation conventions for this work.

| Notation | Meaning |
|---|---|
| $\mathbf{x}$ | Vector. |
| $|\mathbf{x}|$ | Amount of elements in Vector $\mathbf{x}$ |
| $\mathbf{x}[i]$ | Element with index $i$. |
| $\mathbf{x}\langle i, w \rangle$ | Vector defined by window of size $w$ around index $i$ |
| $\mathbf{x}'$ | Min-max normalized vector elements |
| $\dot{\mathbf{x}}$ | First derivative |
| $\bar{\mathbf{x}}$ | Arithmetic mean of all vector elements |
| $\tilde{\mathbf{x}}$ | Savitzky Golay [15] filtered vector elements |
| $\hat{\mathbf{x}}$ | Linear regression model, based on index and elements of $\mathbf{x}$ |
| $s(\mathbf{x}, \mathbf{r})$ | Residual standard deviation |
| $\mathbf{m}$ | Complete measurement with all segments |
| $\mathbf{m}_a, \mathbf{m}_b, \mathbf{m}_c$ | Analyte, baseline and clearing segment |
| $M$ | Set of all measurements of a sensor screening |

### 3.1. Slope-Based Curve Signature

As mentioned in the motivation, one challenge to be solved algorithmically is to provide support for validating the screening measurements. To be as efficient as possible, without having detailed knowledge about the behavior of the sensor itself, the fastest way is to search for similar curves to a given reference time series.

Since the proposed signature is based on the curve's slope, the algorithm works on the first derivative of the Savitzky–Golay [15] filtered measurement time series. It assigns each measurement a sequence comprised of the symbols $+, -$ and $*$, representing its slope characteristic.

In preparation for the signature, a threshold $t$ for the measurement's noise is needed. It is calculated using the residual standard deviation $s$ of the baseline's last 200 s before analyte exposure and multiplying it with a customizable tolerance factor $\tau_v$ according to:

$$t = s(\mathbf{x}, \hat{\mathbf{x}}) \cdot \tau_v, \text{ with } \mathbf{x} = \dot{\tilde{\mathbf{m}}}_b \langle |\dot{\tilde{\mathbf{m}}}_b|, 199 \rangle \tag{3}$$

Each sample from the first derivative is then assigned a symbol as follows:

$$\operatorname{sgn}(\mathbf{m}, i) = \begin{cases} + & \text{if } \dot{\tilde{\mathbf{m}}}_a[i] > t, \\ - & \text{if } \dot{\tilde{\mathbf{m}}}_a[i] < -t, \\ * & \text{else} \end{cases} \tag{4}$$

If the absolute value of $\dot{\tilde{\mathbf{m}}}_a[i]$ is smaller than the noise threshold $t$, it is assigned the $*$ sign, indicating that the slope is caused by noise. Else the sgn function values are coded as either $+$ or $-$. The values assigned with the $*$ symbol are not important to the signature itself, but are needed for correct hysteresis filtering. The resulting symbols from the sgn function are concatenated into a sequence, resulting in the measurement slope signature. The final signature is created, hysteresis filtered and simplified as follows:

- Build `sig` by concatenating results of $\operatorname{sgn}(\mathbf{m}, i)$ for each sample.
- Delete all leading $*$ from `sig`
- Replace each remaining $*$ with the immediately preceding $+$ or $-$ symbol
- Delete all symbols that are not part of an at least $\omega_s$ long sub-string of the same symbol
- Reduce all identical consecutive occurrences of the same symbol to one occurrence

Following is a non-exhaustive list of well-known curve similarity measures that can be extended by the proposed signature.

- Area Method [16]
- Discrete Fréchet Distance [17]

- Partial Curve Mapping (PCM) [17]

We decided to use a simple difference-based approach for our application example, since it is fast and sufficient. This calculation is performed on the analyte segment of the reference curve $\mathbf{r} \in M$ and the curve under test $\mathbf{m} \in M$, where $\mathbf{m_a}'$ represents a min-max normalized analyte segment of a measurement.

$$d(\mathbf{m}, \mathbf{r}) = \frac{\sum_{i=1}^{|\mathbf{m_a}|} |\mathbf{m_a'}[i] - \mathbf{r_a'}[i]|}{|\mathbf{m_a}|} \tag{5}$$

Because all calculations are performed on the min-max normalized curves and the resulting sum is divided by $|\mathbf{m_a}|$, identical curves have a distance of 0, whereas the maximum distance is limited to 1.

### 3.2. Feature Extraction

Before detailed definitions of the actual features are given, the helper function $u$ is introduced. It yields the smallest index of a measurement $\mathbf{m} \in M$, at which the average of a surrounding window reaches a relative amount $\tau_u$ of the reaction's peak. The threshold $\tau_u$ and the window size $\omega_u$ can be chosen as needed. The arithmetic mean of all values in a measurement is denoted with $\bar{\mathbf{m}}$. The definition of $u$ is based on a case differentiation regarding the main direction of the reaction's slope:

$$u(\mathbf{m}, \tau_u) = \begin{cases} \min\{i \mid z \geqslant \tau_u\} & \text{if } \mathbf{m} \text{ has a positive-slope reaction,} \\ \min\{i \mid z \leqslant 1 - \tau_u\} & \text{else,} \end{cases} \tag{6}$$

$$\text{with} \quad z = \overline{\mathbf{m_a'}\langle i, \omega_u \rangle}, \quad i \in \mathbb{N}, 1 \leqslant i \leqslant |\mathbf{m_a}| \quad \text{and} \quad \tau_u \in \mathbb{R}, 0 < \tau_u < 1.$$

Moreover, some of the features depend on the slope $m_{\hat{\mathbf{x}}}$ of a linear regression model, defined in the following sample-wise definition of $\hat{\mathbf{x}}$.

$$\hat{\mathbf{x}}[i] = m_{\hat{\mathbf{x}}} \cdot i + b_{\hat{\mathbf{x}}} \tag{7}$$

The overall performance indicator for each measurement is calculated based on quantifiable features, which are each defined as a function $f_j : M \to \mathbb{R}$. All features are part of the feature set $F$ and can be referenced using an index $j \in \mathbb{N}$ with $1 \leqslant j \leqslant |F|$.

The following initial set of features was identified after interviewing a domain expert in the field of MOX gas sensors. The features were then formalized in the following list. All features marked with $^{-1}$ need to be inverted after normalization because a higher value will always be considered better for the performance metric introduced later in this section.

1. Sensitivity

   The sensitivity quantifies how strong a sensor reacts to the analyte it is exposed to [18,19].

   It is calculated by subtracting the mean-value of a window $a = \overline{\mathbf{m_a}\langle |\mathbf{m_a}|, 119 \rangle}$ containing the samples of the last 120 s of analyte exposure from the mean value of a window $b$ that contains the samples of the last 120 s before gas exposure (baseline), divided by the latter.

$$f_1(\mathbf{m}) = \frac{b-a}{b} \quad \text{with} \quad b = \overline{\mathbf{m_b}\langle |\mathbf{m_b}|, 119 \rangle} \quad \text{and} \quad a = \overline{\mathbf{m_a}\langle |\mathbf{m_a}|, 119 \rangle} \tag{8}$$

2. Reaction Speed I $^{-1}$

   This measure is an indicator of how fast the sensor reacts to the analyte it is exposed to. It covers the time from the start of the exposition to the analyte until the reaction reaches 50 % of its overall strength.

$$f_2(\mathbf{m}) = u(\mathbf{m}, 0.5) \tag{9}$$

3. Reaction Speed II $^{-1}$
   The time between reaching 50% and 90% of the maximum reaction is used as a second measure for the reactivity of the sensor.

   $$f_3(\mathbf{m}) = u(\mathbf{m}, 0.9) - u(\mathbf{m}, 0.5) \tag{10}$$

4. Plateau Quality $^{-1}$
   Ideally, after a transient response, the sensor signal will reach a plateau. Therefore the slope of a linear regression curve between the point where 90% of the maximum signal is reached and the end of the analyte segment can be used to quantify the quality of this plateau.

   $$f_4(\mathbf{m}) = |m_{\widehat{\mathbf{m}_a(l,r)}}| \quad \text{with} \quad l = u(\mathbf{m}, 0.9) \quad \text{and} \quad r = |\mathbf{m}_a| \tag{11}$$

5. Drift $^{-1}$
   The slope of a linear regression curve fitted through the baseline segment shows a possible drift of the sensor resistance. While a small slope might be acceptable, higher drift leads to possible unstable sensor behaviour in the field.

   $$f_5(\mathbf{m}) = |m_{\widehat{\mathbf{m}}_b}| \tag{12}$$

6. Repeatability $^{-1}$
   The similarity between all measurements of the same sensor/analyte combination is an indicator of the repeatability. The average of the curve distances $d$, introduced with Equation (5), is calculated for all possible combinations. The following equation is an example, defining the feature for the three valid measurements $\mathbf{m}, \mathbf{n}, \mathbf{o} \in M$ per sensor/analyte pair.

   $$f_6(\mathbf{m}) = f_6(\mathbf{n}) = f_6(\mathbf{o}) = \frac{d(\mathbf{m}, \mathbf{n}) + d(\mathbf{m}, \mathbf{o}) + d(\mathbf{n}, \mathbf{o})}{3} \tag{13}$$

7. Dynamic Range $^{-1}$
   It is beneficial for the later integration of the read-out electronics that the sensor work in a low dynamic range. Therefore, the span of the analyte segment of the measurement can be extracted as a feature.

   $$f_7(\mathbf{m}) = \max \mathbf{m}_a - \min \mathbf{m}_a \tag{14}$$

8. Power Consumption $^{-1}$
   The MOX sensors contain a heating element that needs to be heated up to a specific temperature. As mentioned before, the operating temperature has a big influence on the response and the power consumption of the sensor. A goal could be to minimize the power consumption by still maintaining a feasible response. The feature is the average of the heater voltage during the entire measurement. Let $\mathbf{v_m}$ be the heater voltage values for measurement $\mathbf{m}$, if available.

   $$f_8(\mathbf{m}) = \bar{\mathbf{v}}_{\mathbf{m}} \tag{15}$$

9. Signal to Noise Ratio (SNR)
   To compare different sensors to each other, the ratio of signal strength to its baseline noise is a good indicator. To obtain the signal strength, the reaction phase of the

measurement is segmented into rolling mean-valued windows of size 41 samples. Depending on the reaction type, the signal strength is then calculated with:

$$p_s(\mathbf{m}) = \begin{cases} \max \mathbf{a} - z & \text{if } \mathbf{m}_a \text{ has a positive-slope reaction,} \\ z - \min \mathbf{a} & \text{else,} \end{cases} \tag{16}$$

with $z = \overline{\mathbf{m}_b \langle |\mathbf{m}_b|, 40 \rangle}$ and $\mathbf{a}[i] = \overline{\mathbf{m}_a \langle i, 20 \rangle}$, $i \in \mathbb{N}$, $1 \leqslant i \leqslant |\mathbf{m}_a|$

Finally, the SNR is calculated as:

$$f_9(\mathbf{m}) = \frac{p_s(\mathbf{m})}{s\left(\mathbf{m}_b, \widehat{\mathbf{m}_b}\right)} \tag{17}$$

Let $\mathbf{m}_k \in M$ be the measurement with the corresponding index $k \in \mathbb{N}$ for which is claimed $1 \leqslant k \leqslant |M|$. With the features defined in this section, a feature vector $\mathbf{g}_j$ for each feature is calculated.

$$\mathbf{g}_j[k] = f_j(\mathbf{m}_k) \tag{18}$$

For further use, the features are min-max normalized and inverted, if needed. The final feature vector $\mathbf{f}_j$ for each feature is defined as:

$$\mathbf{f}_j[k] = \begin{cases} \mathbf{g}'_j[k] & \text{if feature } j \text{ does not need to be inverted,} \\ 1 - \mathbf{g}'_j[k] & \text{else.} \end{cases} \tag{19}$$

### 3.3. Quantifiable and Individually Prioritizable Ranking Metric

To rank the sensors according to the selected features, an overall performance value for each measurement is calculated with

$$\mathbf{p}[k] = \prod_{j=1}^{|F|} p_j\left(\mathbf{f}_j[k]\right) \tag{20}$$

and the linear feature-specific priority function $p_j(x)$

$$p_j(x) = \phi_j \cdot x - \phi_j + 1 \quad \text{with} \quad \phi_j \in \mathbb{R}, \, 0 \leqslant \phi_j \leqslant 1 \tag{21}$$

where the priority value $\phi_j$ can be chosen by the user for each feature to prioritize it individually during the calculation. To simplify things, we specified a set of five priority values, resembling the following priority levels:

$$
\begin{aligned}
\text{Lowest}: \quad & \phi_j = 0.1 \\
\text{Lower}: \quad & \phi_j = 0.3 \\
\text{Normal}: \quad & \phi_j = 0.5 \\
\text{Higher}: \quad & \phi_j = 0.7 \\
\text{Highest}: \quad & \phi_j = 0.9
\end{aligned}
$$

Figure 2 shows the influence of $\phi_j$ for these predefined levels. It is important to understand how the priority value steers the influence of a feature within the performance indicator. With each feature value $\mathbf{f}_j[k] < 1$ involved in the product, the performance indicator $\mathbf{p}[k]$ for measurement $\mathbf{m}_k$ will decrease. This demotion capability is restricted by $p_j$ to $1 - \phi_j \leqslant p_j(\mathbf{f}_j[k]) \leqslant 1$. Hence, a lower $\phi_j$ will give the feature a lower priority compared to features with a higher $\phi_j$ and vice versa.

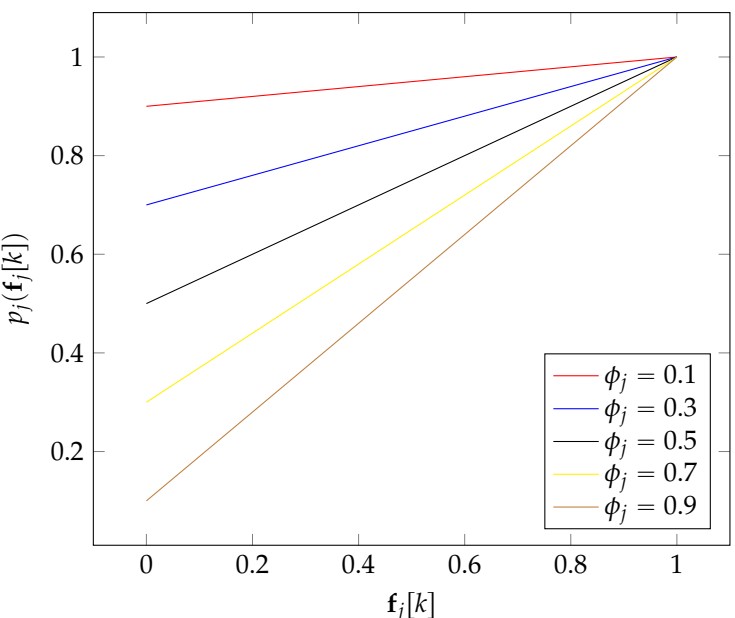

**Figure 2.** Graphs of boosting function $p_j$ for different $\phi_j$.

Consider, for example, the worst measurement $\mathbf{m}_w$ for feature $j$ with $\mathbf{f}_j[w] = 0$; then, setting $\phi_j = 1$ demonstrates the feature's full demotion influence on the performance indicator by annihilating $\mathbf{p}[w]$ completely:

$$p_j(\mathbf{f}_j[w]) = 0 \quad \Rightarrow \quad \mathbf{p}[w] = 0 \tag{22}$$

By selecting $\phi_j = 0.5$ instead, $\mathbf{f}_j[w]$ is now only capable of decreasing the performance indicator for $\mathbf{m}_w$ to 0.5.

The min-max normalized performance indicator $\mathbf{p}'$ now holds the respective performance value for each measurement, where $\mathbf{p}'[x] = 1$ applies to the best overall performing measurement $\mathbf{m}_x$ for the selected feature set. The final ranking of the measurements can be achieved by sorting $\mathbf{p}'$.

## 4. Application Example, Results and Discussion

An important design target for mobile applications is to minimize power consumption. Because MOX gas sensors utilize a significant amount of power for heating their sensitive layer to a suitable working temperature, researchers are continuously trying to optimize substrate compositions that do not require high operating temperatures while still performing adequately for a specific application. In the following, we will therefore illustrate the suitability of the proposed algorithms for finding the most energy-efficient sensor for Acetone detection based on the data of the sensor screening described in Section 2.1. Initially, the software which was developed for this work will be briefly introduced as the platform used for the application example.

### 4.1. Software

To support a user in all tasks related to data processing and evaluation, graphic user interface (GUI) software, depicted in Figure 3, was developed. To display and navigate through the data, the GUI implements a tree based navigation with filtering functionality that is always visible on the left side of the software. To sort and structure the data, the measurements are hierarchically arranged top-down starting with the analytes, followed by the virtual sensors which group the associated measurements for the specific combination together.

The software is divided into tabs, according to the introduced algorithms: View + Manual Validation, Auto Validation and Sensor Ranking. Each tab encapsulates the controls and

views needed for the respective use case. Depending on the active tab, the navigation is either used to browse through all measurements, choose a reference curve for the similarity algorithms or select the combinations of sensors and analytes for the ranking.

The filter enables the user to specify the following parameters:

- Analyte
- Sensor Substrate
- Sinter Temperature
- Sinter Time
- Sensor Operating Temperature
- Validation Status

Furthermore, the user can add measurements to a list of favorites or use the reference checkbox to obtain a list of all measurements that have been used as references in the curve similarity algorithm.

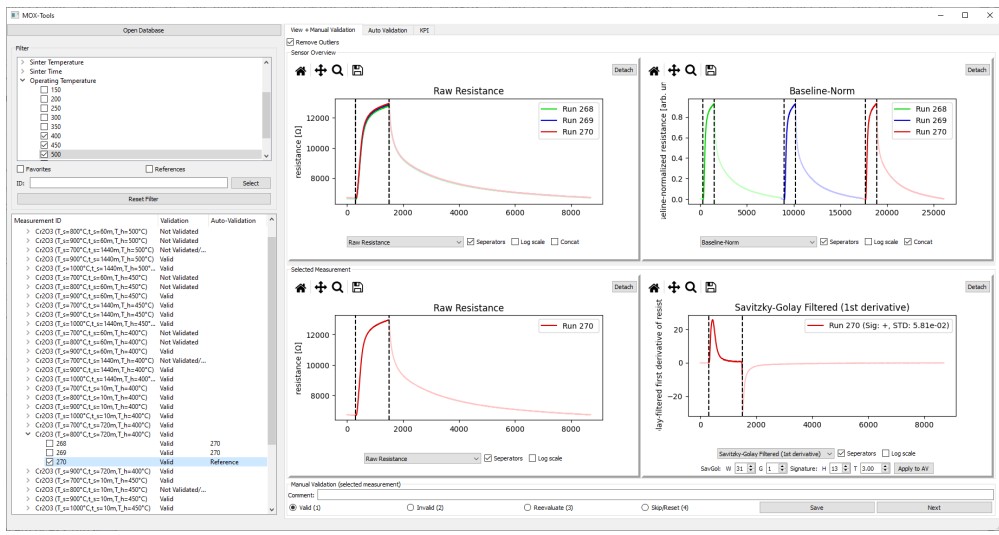

**Figure 3.** The software showing the View + Manual Validation tab.

Visual inspection is realized with four interactive graph views divided into two subgroups. The upper graphs are used to display all measurements for the selected combination of virtual sensor and analyte, whereas the lower ones show the specific measurement selected in the navigation tree. The user is able to inspect the data by applying several filters and standardizations (e.g., first derivative, baseline normalized resistance, etc.).

After an in-depth inspection, a validity status can be assigned to the measurement manually by the user. A measurement can have three validation states:

- *Valid*
- *Invalid*
- *Not Validated*

All measurements are initially in the *Not Validated* state. Manual validation and annotation is realized with four numbered radio buttons, a commentary field and two buttons. Remarks and textual annotations can be added to the Comment text field. In addition to the mentioned states, *Reevaluate* marks the measurement for later inspection, whereas *Skip/Reset* either resets its validation state back to *Not Validated* or skips the measurement if it is *Not Validated*. Care was taken to minimize the amount of clicks by adding keyboard shortcuts and effective tabbing. Using the shortcuts, the validation and textual comments are saved and the software automatically navigates to the next curve for inspection without any needed mouse interaction.

### 4.2. Automatic Measurement Validation

The first step before the measurements can be ranked is to remove those without useful information. This is done automatically as mentioned before by using established curve similarity metrics in conjunction with the presented slope-based signature calculation algorithm. The software implements several similarity methods, all of which compare two time series to each other. The complete functionality is encapsulated in a separate software tab and depicted in Figure 4.

All metrics calculate and assign a score to each measurement and afterwards rank them with descending similarity in the middle list (yellow rectangle in Figure 4). The user can inspect candidate curves and reference together in an interactive graph view (purple) and afterwards apply the final validation with the buttons and the following list selection: The measurements moved to the upper list (red) are set to invalid, the status of those in the middle list are not changed and finally the lower list (green) marks its contents as valid. The option *Remove low SNR* (blue) automatically proposes measurements as invalid that do not show enough signal amplitude by calculating the measurements signal to noise ratio (SNR) and comparing it to the threshold given in the spin box.

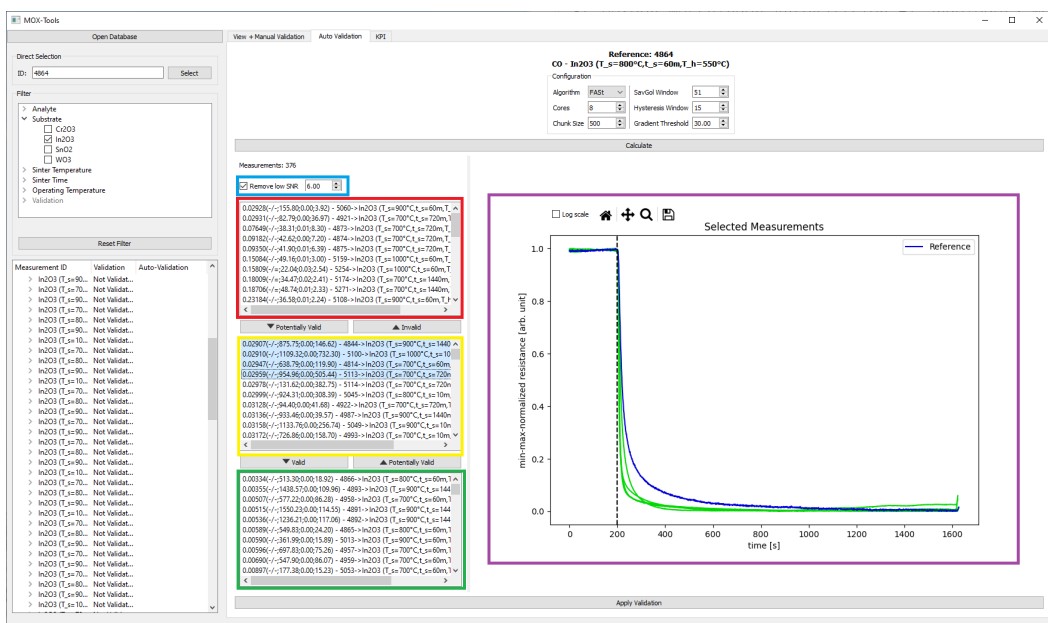

**Figure 4.** Exemplary use of the auto validation function. Validation of all CO measurements performed with $In_2O_3$-based sensors. The blue curve is the user-supplied reference, the green curves are the candidates as selected in the middle (yellow rectangle) and lower list (green rectangle). The upper list (red rectangle) holds all measurements that are greater than the selected reference threshold (light blue rectangle) and are therefore sorted out.

For validation of the proposed signature algorithm, we created a test subset including the runs 52, 261, 267, 343 and 374 and calculated the distance with respect to the reference run 270 for all available curve distance methods. The proposed slope-based signature algorithm yields the same signature + for the reference and all runs of the subset except for run 52, which was assigned the signature $-+$. In Figure 5 run 52 shows a significant drop and therefore a different slope characteristic compared to the other runs, which is represented by the signature value. Referring to Table 3, the calculated curve similarities based on the four curve distance methods listed in Section 3.1 reveal that run 52 has a very similar distance to the reference compared to at least one of the other runs for the respective method. If the signature would not be used to sort out run 52, it would be on the same similarity level as the other curves, ignoring the significant difference in slope characteristics.

**Table 3.** Distance values for the four implemented distance metrics for the runs in the selected test set, compared to the reference run 270.

| Run | Method | Distance to 270 | Distance of 52 to 270 |
|-----|--------|-----------------|------------------------|
| 261 | PCM | 68.82963 | 68.62570 |
| 267 | Area | 268.05069 | 264.02023 |
| 343 | Fréchet | 0.62149 | 0.62126 |
| 374 | Point | 0.23268 | 0.22901 |

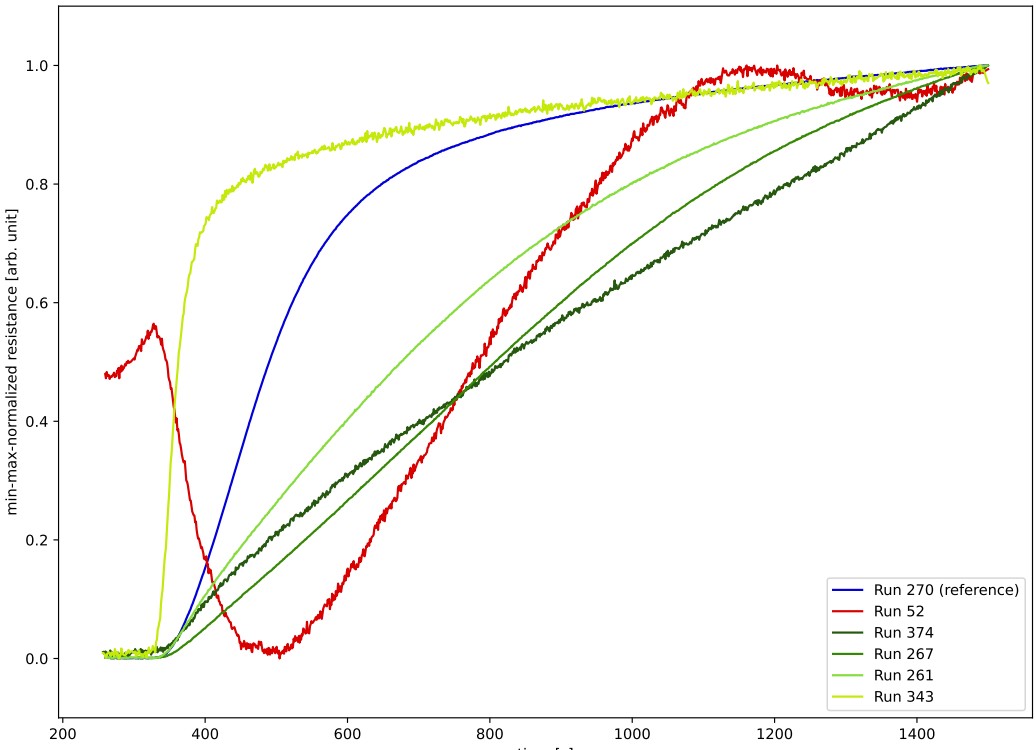

**Figure 5.** Curve similarity of a subset of test curves with respect to the reference run 270 (blue). The curve of run 54 (red) yields a different signature $(-+)$ to the other curves (green) $(+)$ which is therefore filtered out although it has very similar distances (see Table 3) to the remaining curves.

*4.3. Features and Raking*

After the measurements are validated, the scenarios question needs the following features from the set introduced in Section 3.2 to find the most power-efficient sensor: *Power*, *Sensitivity*, *Reaction Speed I*, *Reaction Speed II* and *Repeatability*. The priorities were set as listed in Table 4.

**Table 4.** Priorities of the features used for the exemplary ranking.

| Feature | | Priority | | Comment |
|---------|--|----------|--|---------|
| Power | $(f_8)$ | Highest | $(\phi_8 = 0.9)$ | Cooler sensors need less power. |
| Sensitivity | $(f_1)$ | High | $(\phi_1 = 0.7)$ | Better for small amounts of the gas. |
| Reaction Speed I | $(f_2)$ | Normal | $(\phi_2 = 0.5)$ | Hot sensors have higher speeds. |
| Reaction Speed II | $(f_3)$ | Normal | $(\phi_3 = 0.5)$ | $\phi_3, \phi_2 = 0.5$ are a good trade of. |
| Repeatability | $(f_5)$ | Normal | $(\phi_5 = 0.5)$ | Consider sensor stability ov. time. |

The resulting list in Table 5 shows the most power-efficient sensor for the task of measuring Acetone and is depicted in Figure 6. The second and third best sensors are

shown in Figure 7. Furthermore an exemplary midfield sensor and the worst sensor from the ranking can be found in Figure 8.

**Table 5.** The top 3, midfield and worst sensors from the available validated dataset ranked according to the most power-efficient (coldest sensor operation) detection of the analyte Acetone.

| Rank | $p'$ | Substrate | Sinter Temp. (°C) | Sinter Time (minutes) | Op. Temp. (°C) | Sensitivity (Arb. Units) |
|---|---|---|---|---|---|---|
| 1 | 1 | $Cr_2O_3$ | 1000 | 1140 | 350 | 0.77 |
| 2 | 0.95 | $Cr_2O_3$ | 900 | 10 | 400 | 0.79 |
| 3 | 0.86 | $Cr_2O_3$ | 1000 | 1440 | 400 | 0.66 |
| . . . | . . . | . . . | . . . | . . . | . . . | . . . |
| 240 | 0.4 | $Cr_2O_3$ | 700 | 720 | 350 | 0.48 |
| . . . | . . . | . . . | . . . | . . . | . . . | . . . |
| 480 | 0 | $Cr_2O_3$ | 1000 | 10 | 550 | 0.18 |

The first three sensors are very similar concerning their sensitivity (approximately 0.7) and reaction speed toward the analyte as shown in the baseline-normalized depiction in Figures 6 and 7. Yet, the performance value of the sensor with the smallest power consumption of these three was chosen to be first due to the selected feature prioritization. To put the best sensor into perspective, a sensor from the midfield and the worst performing sensor of the ranking are depicted in Figure 8. While the midfield sensor is operated at the same temperature as the best sensor, it is demoted due to its lower sensitivity towards Acetone of only 0.48. The last and therefore worst sensor in the ranking delivers a much lower sensitivity of just 0.18 whilst consuming more power to operate at the higher temperature of 550 °C. It is therefore the least favorable choice for this particular scenario.

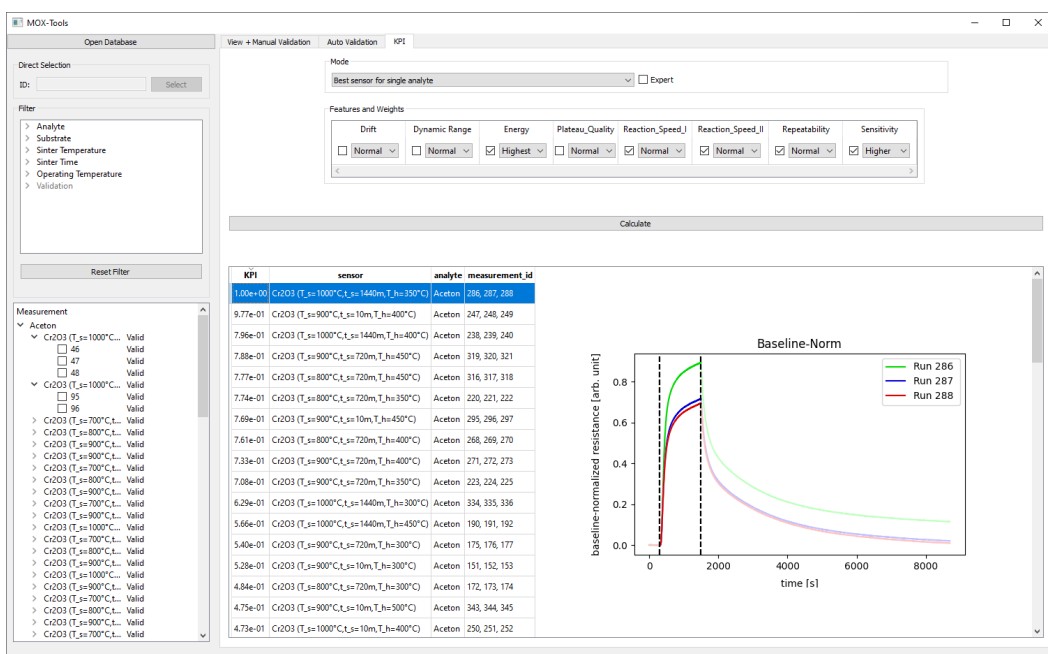

**Figure 6.** The final ranking, showing the measurement belonging to the best sensor for the given problem.

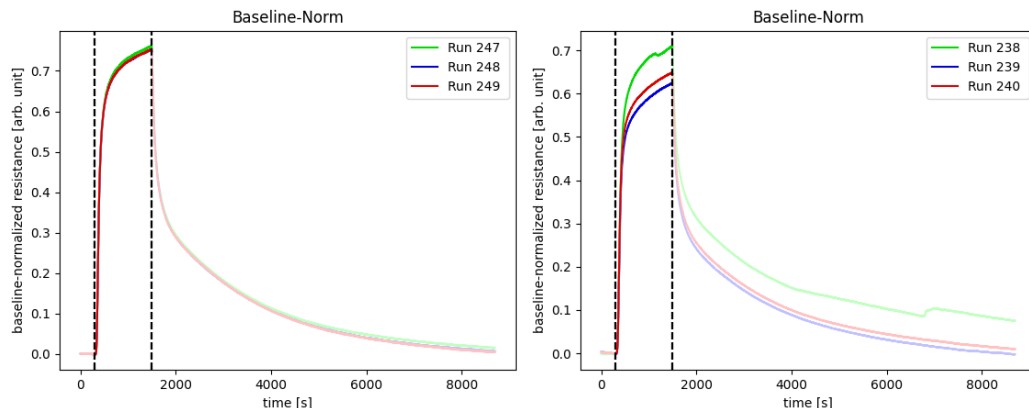

**Figure 7.** Second and third place of the final ranking.

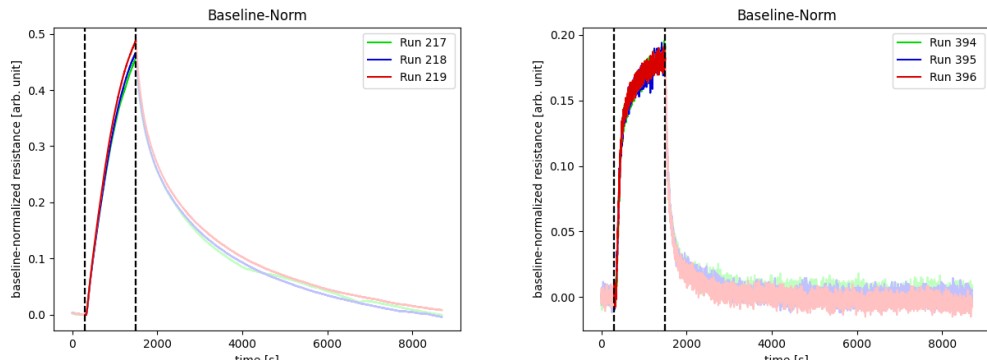

**Figure 8.** Midfield place of worst sensor in the ranking.

## 5. Conclusions

In this work algorithms for validating measurements and a feature-based sensor ranking have been presented. To address the challenge of automatic validation of the extensive screening data, a slope-based signature calculation has been proposed as an addition to established curve similarity metrics. Using the newly presented signature-extraction algorithm, curves that differ in slope (shape) are now much more clearly separated, which directly leads to much faster post-processing time for the measurement validation. For the other major challenge, a sensor performance ranking, a set of features and a ranking metric have been introduced. The features, obtained by interviews with experts in the domain of gas sensor screening, were first of all mathematically formalized and afterwards algorithms were implemented to extract and optionally normalize quantifiable information from the time series. The performance metric offers individual prioritization of features and allows to rank the measurements according to their overall performance on all features used.

Finally, the proposed algorithms were used to validate and rank various sensors in a large data set obtained during an extensive screening. It was shown that the additional use of the proposed slope-based signature delivers better results compared to the established curve distance methods that do not take slope characteristics into account. This new algorithm combination can help validate many measurements more efficiently. The ranking and feature extraction algorithms were tested by taking on the question of which sensor has the highest sensitivity towards a specific analyte under low-power constraints. A prioritization method for the quantifiable features was developed and implemented to be able to adapt the ranking to multiple scenarios of interest.

The software suite implemented for this work can be used as a solid foundation for future measurement campaigns, as it provides not only an extensible feature extraction, but also offers a structured storage model and can be used as a general management platform for screening data. Future goals are improving the outlier detection, extending

and refining the current feature set and integrating the control and acquisition protocols for the automatized sensor screening into the software suite.

**Author Contributions:** C.H. is the lead author and was responsible for the data processing, software conception, the semi-automatic curve similarity implementation and writing of the final paper. S.S. is co-author of this work. He implemented and conceptualized the software and co-wrote the paper. J.W. designed and performed the measurements, he was interviewed as expert for the implemented feature set. P.K. and N.J. are the directors of the Institute of Safety and Security Research (ISF). P.K. was also interviewed as expert for the features. R.T. and N.J. are the referees in the PhD proceedings of C.H. All Authors contributed by providing advice, experimental guidance, project coordination and iterations of paper review. All authors have read and agreed to the published version of the manuscript.

**Funding:** This research received internal funding from the Institute of Safety and Security Research (ISF) at BRS-U.

**Institutional Review Board Statement:** Not applicable.

**Informed Consent Statement:** Not applicable.

**Data Availability Statement:** Not applicable.

**Conflicts of Interest:** The authors declare no conflict of interest.

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
