# Peer review of "Algorithms for Automatic Data Validation and Performance Assessment of MOX Gas Sensor Data Using Time Series Analysis"

_algorithms, doi:10.3390/a15100360_

Round 1
Reviewer 1 Report
The introduction is not concrete. The authors pointed out the challenges but have not included the point-to-point solutions proposed to address the challenge and stated why they should be a good solution. Those narratives are important to point to the novelty of this research. Additionally, more others’ works need to be cited to provide a comprehensive state-of-the-art review.
Something unclear in the mathematical section with a list are given here:
1. X<I,w>: the definition of the mathematical operator is unclear. A specific instance should be given to explain what it means.
2. Equations should be numbered
3. In the equation below line 107, how the “199” comes from? Starting from 0 s and every increment is 1 s till to 200 s? It is very confusing and needs to be clarified in the text. The authors mentioned this equation stands for a residual standard deviation, which might be a well-standard metric. A relevant citation is needed.
4. Line 133, the metric sensitivity normally means the change of the output divided by the change of the input. How is the current sensitivity form related to the normal definition? also, the citations for those metrics should be given as those should be the standard definitions.
5. how were the priority numbers in Table 3 determined? How sensitive the ranking is to variable priority numbers. For example, change the power priority to 1, and sensitivity priority to 0.8 or 0.6. These will affect the ranking, so how have current numbers been determined?
6. More information in Table 4 should be provided for readers' interest. A direct comparison or summary of how sinter temp. and time and operation temp affect the sensitivity would be interesting to see.
Author Response
Dear respected Reviewer,
thank you very much appreciated input. We have addressed your comments and updated the manuscript accordingly:
Indeed, the introduction was missing references to closely related work. We have looked at various publications in the field of MOX-Sensor screenings and found out, that the actual processing of data evaluation/validation is mostly not covered. Many publications describe the used methods and feature crudely or even as black-boxes entirely. The few related works, have now been added to the manuscript on lines 65 to 81. The absence of a well-described solution for screening evaluation with clear and formally defined algorithms and feature extraction is the main reason, why we decided to write our paper in the first place and why we took great care to mathematically and formally define the features and the processes and methods that we used to validate and rank our sensors.
The introduction was very abrupt. We've added the ideas and concepts for the solutions for the challenges as a preview for what follows in the work. These changes can be found in Lines 47 to 63.
As for the remaining remarks:
1.) We've added a concrete example in lines 130 - 132 to better explain how the window function works. This should hopefully also directly remedy the point No. 3. We hope that with the new explanation this is no longer confusing.
2.) Equations are now numbered.
3.) The formula for the residual standard deviation has been added as a reference to the work. The window function has been described in further detail, along with a concrete example, of how to use it.
4.) The information was given to us during the interviews with our domain experts and described as the way how the sensitivity is calculated for MOX Gas sensors. We have additionally verified this and added two new citations.
5.) We wanted to show how the process of screening-evaluation benefits from automated feature extraction and ranking. Therefore, we choose the research question of power efficiency as possible example. We ourselves chose the summarized values in Table 3 in a way we seemed fitting, for the exemplary scenario. Since we want a sensor that is very sensitive at the lowest possible power consumption. We prioritized the features therefore as high (sensitivity) and highest (power).
The results directly reflect this. There is actually a sensor that is slightly better in terms of sensitivity, but requires more power and is therefore degraded to 2nd place (see table 4). In further discussions with our domain experts, they stated that the reaction speed and the repeatability (to avoid a lucky shot) should also feature in this rank, but only with normal priority.
6.) As this work's focus is on the algorithms and validation methods to differentiate between well known and unknown measurement results, feature extraction and ranking we did not draw any contextual conclusions form the manufactured sensors. We would kindly refer to an upcoming work by our co-author Warmer, in which the contextual performance of the sensors regarding their production parameters and sensitivity are discussed in depth.
Thank you and best regards
Christof Hammer

Reviewer 2 Report
The method proposed by the author is novel, but great improvements are still needed in the manuscript.
1. In the introduction, it is necessary for the author to review recent research in this field.
2. The difficulties of automatic data verification proposed by the author and the advanced techniques of this manuscript should be mentioned in the introduction.
3. Incorrect format in Table 4 and Figure 4.
4. The author needs to express the shortcomings of the method in a pertinent way in the conclusion so that readers can have a clearer understanding
Author Response
Dear respected Reviewer,
1.) Indeed, the introduction was missing references to closely related work. We've looked at various publications in the field of MOX-Gas sensor screenings and found, that the actual processing of data evaluation/validation was mostly not covered. Many publications describe the used methods and features crudely or entirely as black boxes.
The few related works, have now been added to the manuscript in the lines 65 to 81. The absence of a solution for screening evaluation with clear and formally defined algorithms and feature-extraction is the main reason, why we decided to write our paper and why we took great care to mathematically and formally define the features and the processes and methods, that we used to validate and rank our sensors.
2.) Thank you for your input. We have added two paragraphs in which we give a teaser-preview of what's to come later on the work. Indeed the ending of the introduction was very abrupt and missing this. The introduction was very abrupt. We therefore added the ideas and concepts for the solutions for the challenges as a tease for what follows in the work. These changes can be found in Lines 47 to 63.
3.) Thank you. We corrected the format accordingly.
4.) This comment is a bit unclear to us. Do you want to report about shortcomings, that we have with our algorithms, or did you mean to better describe the outcome instead?
If so, we have added more detail to the conclusion on what the work brought forward. Furthermore, allow me to state that the software will be published for the community, once "licensing and code-optimization" has been done. Additionally our co-author Johannes Warmer will be publishing a work soon, in which the actual manufacturing and contextual sensors response will be evaluated in depth. He will be using the algorithms especially for this purpose.
Thank you and best regards
Christof Hammer

Reviewer 3 Report
Review of the manuscript algorithms-1913753 ‘Algorithms for Automatic Data Validation and Performance Assessment of MOX Gas Sensor Data using Time Series Analysis’ by Hammer et al. submitted to Algorithms.
Recommendation: ACCEPT.
Focus of paper: The improvement of the methods of MOX gas sensor data detection. To this end, Hammer et al. presented the developed algorithms for validating measurements and a feature-based sensor ranking for detecting gas using time series analysis. The authors shown that the use of their novel slope-based signature shows better results compared to the existing curve distance methods, because it considers slope characteristics. Moreover, Hammer et al. demonstrated that their new algorithms can help validate measurements more efficiently. They tested ranking and feature extraction algorithms by considering which sensor has the highest sensitivity towards a specific analyte under low-power constrains.
Abstract is well written and clearly describes the undertaken study.
Structure: The paper is well organized with structured sections. The structure of the article conforms to an acceptable format of standard sections: Introduction, Methodology, Results, Discussion, Conclusion, References. Some sections are divided into the minor subsections for a better structure. Logic and numeration of the sections is correct and consecutive.
Introduction presents a background, defines research goals and provides a clear statement of research problem. The Introduction well describes the research and introduces the current state-of-the-art situation. Hammer et al. performed the review regarding the existing sensor screening methods using 64 sensors exposed to 9 different analytes. These are different in substrate composition, sinter times and sinter temperatures
Research goal and objectives are identified: Hammer et al. aimed to develop algorithms for semi-automatic validation, feature extraction, and ranking of time series measurements acquired from MOX gas sensors.
Literature regarding the relevant topics is reviewed, formatted according to the journal rules and appropriately referenced. Hammer et al. reviewed the use of MOX gas sensors and discussed their advantages. Major sources include published papers on Electronics, Material Technologies and Sensors. The authors cited 12 papers.
Research gaps and weakness in former works are described; the existing gaps are identified. The contribution of this work filling this gap is explained. It concerns the developing of the algorithms for validating measurements and a feature based sensor ranking for MOX gas sensors using automated approach.
English language: acceptable.
Data used in this study are described: Hammer et al. used the data set collected during an extensive screening consisting of 16,320 measurements, all taken with different sensors at various temperatures and analytes. Data are explained, sources are mentioned.
Methods used in the study are summarized: Methods include developed algorithm for data analysis from MOX gas sensors. Hammer et al. performed semi-automatic measurement validation by extending established curve similarity algorithms with a slope-based signature calculation. Also, the authors used the feature-based ranking metric for individual prioritization of each feature to find the best performing sensors. The methods are descriptive. Modifications of the existing algorithms and methods are mentioned briefly. The study well explained the methods. Hammer et al. used the 2 approaches: 1) Slope-based signature calculation as a curve similarity metric; 2) Feature-based ranking metric to sort the sensors according to performance. The methodology is structured and clearly described. It includes the sufficient information to reproduce methods in a similar research, which is useful for future studies.
Motivation and research gaps are explained: The problem of effective screening of gas which is normally a time-consuming task (i. e., several days) and needs automatization. Therefore, the acquired data has to be analysed and interpreted to achieve the final goal of finding the best fitting sensor and its optimal operating temperature for a given target. Hammer et al. noted that the metal-oxide (MOX) sensors can detect gas concentrations to required level, but are not selective enough. Therefore, there is a need to create and test new material combinations with the goal of building sensors that are selective and sensitive to gas using the advanced identification parameters. Another problem is the large amount of raw data which requires automatization of algorithm. Finally, another goal was to to optimize substrate compositions, that don’t require high operating temperatures, thus, to minimize power consumption.
Results are reported: Hammer et al. prepared the dataset of time series which consisted of three segments and durations, using 64 sensors. The authors described the workflow on data acquisition. Thus, they varied operating temperatures during screening, to record the resulting impact on the sensors performance. Hammer et al. sampled sensor resistance, heater voltage and heater current with 1 Hz. The single measurements for a sensor and an analyte were repeated. The data were preprocessed using the out of range (OOR) detection algorithm which checks the threshold for the data. The authors analysed deviations, checked continuous current flow and validated the resistance characteristic of the integrated Platinum heater element of the sensors at different temperatures. Afterwords, they used the algorithms for validation and ranking. The formal conventions in proposed algorithms are introduced and parameters explained. Hammer et al. used the slope-based curve signature to validate the screening measurements, which works on the first derivative in filtered measurement time series. The techniques is described with presented equations. Hammer et al. used a simple difference-based approach for their application example, since it is fast and sufficient. The feature extraction was performed and explained in detailed in 9 working steps. The quantifiable and Individually ranking metric was used to rank the sensors according to the selected features and 5 prioritised values, ranked from the lowest to the highest with sub-classes. The performance of boosting function is illustrated. The overall performance indicator for each measurement is calculated based on quantifiable features and presented in 9 described steps. The automatic measurement validation was based on the comparison of the two time series to each other. Thus, Hammer et al. calculated all metrics and assigned a score to each measurement and afterwards ranked them with descending similarity in the middle list.
The Results are presented with clarity and include description, graphical illustrations, tables, equations and calculations. Results are relevant to the initially defined research goals and objectives. The results highlights the major achievements of this study.
Discussion interpreted the major outcomes of this study. The advantages of the obtained results are described and compared with other studies. The discussion described the issues of methodology and results. Thus, the functionality of the algorithms, is demonstrated and commented. The developed software suite is illustrated with an exemplary scenario, showing how to find the most power-efficient MOX gas sensor in a data set.
Conclusion summarized the study with interpretation of facts. The importance of this paper is well summarized as follows: Hammer et al. improved the two identified main tasks for the automatized data processing in gas screening. This includes validation and ranking aimed at evaluating the performance of screening and identification of the best sensor based on quantified performance metrics. The conclusions are appropriately stated and connected to the original questions.
Actuality, novelty and importance of the research is clear. It consists in technical approach for semi-automatic validation, feature extraction, and ranking of time series measurements acquired from MOX gas sensors. Hammer et al. developed algorithms for finding the most energy-efficient sensor for Acetone detection based on the data of the sensor screening and presented the novel algorithm which works effectively.
Logic: The clarity of the text logic and organization of the paper is sufficient. It demonstrates the consistent interpretation of the results with detailed explanations. A comparison of the results with previous studies is presented.
Relevance: The manuscript meets general criteria of the significance in algorithms development. Specifically, Hammer et al. presented novel methods of automatic validation of the extensive screening data, a slope-based signature calculation, which has been proposed as addition to the established curve similarity metrics. The study has been conducted in accordance to the technical standards in mathematical and programming science. It is relevant to the journal topic as corresponding to the major clusters and research sub-disciplines: time series analysis, gas sensors, slope based signature, automatic measurement validation, prioritisation, ranking, feature extraction.
Academic contribution is clear: the paper increases the knowledge in algorithms and development of novel solutions in optimization. Specifically, Hammer et al. proposed algorithms which were used to validate and rank sensors on a large data set obtained during screening. They shown that the use of their novel slope-based signature shows better results compared to the existing curve distance methods, because it considers slope characteristics. Moreover, the authors demonstrated that their new algorithms can help validate measurements more efficiently. They tested ranking and feature extraction algorithms by considering which sensor has the highest sensitivity towards a specific analyte under low-power constrains. The paper combines technical, mathematical and applied science approaches which presents a multi-disciplinary study well deserved to be published in Algorithms.
Figures are of good quality, easy to read, relevant and suitable, well illustrate the results, relevant to the content, have sufficient resolution, appropriately described and labeled. Especially, figures containing software boosting function, print screens, auto validation function, curve similarity and ranking cases are very illustrative and well done.
Recommendation: This manuscript can be ACCEPTED based on the detailed report above.
With kind regards,
- Anonymous Reviewer.
02.09.2022.

Author Response

(The authors gave the same response as above.)

Round 2
Reviewer 1 Report
Accept
Reviewer 2 Report
The changes to the manuscript meet the reviewers’ requests. So, I recommend the manuscript be accepted for publication.